# Trehalose Production Using Three Extracellular Enzymes Produced via One-Step Fermentation of an Engineered *Bacillus subtilis* Strain

**DOI:** 10.3390/bioengineering10080977

**Published:** 2023-08-18

**Authors:** Xi Sun, Jun Yang, Xiaoping Fu, Xingya Zhao, Jie Zhen, Hui Song, Jianyong Xu, Hongchen Zheng, Wenqin Bai

**Affiliations:** 1College of Biological Engineering, Tianjin Agricultural University, Tianjin 300384, China; sunxi@tjau.edu.cn (X.S.); yangjun@tib.cas.cn (J.Y.); 2National Center of Technology Innovation for Synthetic Biology, Tianjin Institute of Industrial Biotechnology, Chinese Academy of Sciences, Tianjin 300308, China; fu_xp@tib.cas.cn (X.F.); song_h@tib.cas.cn (H.S.); xu_jy@tib.cas.cn (J.X.); 3Key Laboratory of Engineering Biology for Low-Carbon Manufacturing, Tianjin Institute of Industrial Biotechnology, Chinese Academy of Sciences, Tianjin 300308, China; 4Industrial Enzymes National Engineering Research Center, Tianjin Institute of Industrial Biotechnology, Chinese Academy of Sciences, Tianjin 300308, China; zhao_xy@tib.cas.cn (X.Z.); zhen_j@tib.cas.cn (J.Z.); 5Tianjin Key Laboratory for Industrial Biological Systems and Bioprocessing Engineering, Tianjin Institute of Industrial Biotechnology, Chinese Academy of Sciences, Tianjin 300308, China

**Keywords:** trehalose, *Bacillus subtilis*, pullulanase, maltodextrin, malto-oligosyltrehalose synthase, malto-oligosyltrehalose trehalohydrolase, trehalose conversion rate

## Abstract

At present, the double-enzyme catalyzed method using maltooligosyltrehalose synthase (MTSase) and maltooligosyltrehalose trehalohydrolase (MTHase) is the mainstream technology for industrial trehalose production. However, MTSase and MTHase are prepared mainly using the heterologous expression in the engineered *Escherichia coli* strains so far. In this study, we first proved that the addition of 3 U/g neutral pullulanase PulA could enhance the trehalose conversion rate by 2.46 times in the double-enzyme catalyzed system. Then, a CBM68 domain was used to successfully assist the secretory expression of MTSase and MTHase from *Arthrobacter ramosus* S34 in *Bacillus subtilis* SCK6. At the basis, an engineered strain *B. subtilis* PSH02 (*amyE*::*pulA*/pHT43-C68-ARS/pMC68-ARH), which co-expressed MTSase, MTHase, and PulA, was constructed. After the 24 h fermentation of *B. subtilis* PSH02, the optimum ratio of the extracellular multi-enzymes was obtained to make the highest trehalose conversion rate of 80% from 100 g/L maltodextrin. The high passage stability and multi-enzyme preservation stability made *B. subtilis* PSH02 an excellent industrial production strain. Moreover, trehalose production using these extracellular enzymes produced via the one-step fermentation of *B. subtilis* PSH02 would greatly simplify the procedure for multi-enzyme preparation and be expected to reduce production costs.

## 1. Introduction

Trehalose is a stable and non-reducing disaccharide that is composed of two glucose units linked with an α -1, 1-glycosidic bond [1,2]. This nontoxic carbohydrate is initially considered to primarily act as a stored energy source, similar to glycogen [3]. However, over the years, it has been revealed that trehalose may also perform specialized biological functions such as protecting proteins and membranes against various stresses [4,5,6]. In recent decades, trehalose has sparked growing interest in a variety of industrial applications, which are mainly attributed to its structural features and its special role in molecular stabilization [4,7,8]. It has been widely used in pharmaceuticals, foodstuffs, cosmetics, and agricultural products [2,4,7]. For example, the ability of trehalose to protect a broad array of biological materials, such as DNA, proteins, cell lines, and tissues, has opened up possibilities in the pharmaceutical and biotechnology industries [7]. Additionally, studies have proposed that trehalose, as a bioactive nutrient, is a potential tool for regulating blood sugar levels and alleviating diabetes symptoms. It is used as an artificial sweetener and has sweetness equivalent to 40–45% of that of sucrose [2,7]. 

So far, there are two main enzymatic processes that can be used for trehalose production. The original enzymatic process is applying maltooligosyltrehalose synthase (MTSase, E.C. 5.4.99.15) and maltooligosyltrehalose trehalohydrolase (MTHase, E.C. 3.2.1.141) for the production of trehalose from maltodextrins or starch, as shown in a schematic diagram Appendix A [9,10,11,12]. Currently, trehalose synthase (TreS, E.C. 5.4.99.16) is used to produce trehalose from maltose [9,13]. The greatest advantage of this one-step process, through the intermolecular rearrangement mechanism, is simplicity [9,14]. Most studies on trehalose production via TreS have focused on screening and modifying TreS. Several recombinant TreS genes have been cloned from different bacterial strains and heterologously expressed in different hosts [15,16,17,18,19]. However, until now, the low expression level of TreS and the relatively high cost of the maltose substrate were the main reasons for making this approach economically unattractive [9]. The two-enzyme method is still the mainstream technology for industrial trehalose production. Because the main substrate of this process is maltodextrins or starch, the additions of amylase, pullulanase, and glucoamylase are usually used for improving the trehalose conversion rate. Pullulanase, as a debranching enzyme that explicitly cleaves α -1, 6 glycosidic bonds, has been widely recognized in increasing trehalose conversion rate, as shown in Appendix A [20]. However, commercial pullulanases are usually acidic enzymes that do not match the optimum reaction pH of the neutral MTSase and MTHase. 

Although trehalose production has been industrialized by far, there are still some requirements for producing host strains as the main producing strain is *E. coli*, which contains endotoxin and is unfavorable for safe and high-quality trehalose production [10,11,12]. Aimed at this, we hired the safe strain *B. subtilis* approved by the FDA (Food and Drug Administration) as the host and expressed heterologous enzymes MTSase and MTHase in strain *B.subtilis* SCK6 [2,21]. This work was thought to establish the basis of high-quality trehalose production with low-endotoxin content. In this present work, a neutral pullulanase PulA, which has been developed with high specific activity, was used to enhance the trehalose conversion rate in the two-enzyme trehalose production method. MTSase and MTHase from different bacterial strains have been heterologously expressed and screened for conversion activity. On this basis, a series of engineered strains, co-expressing the two or three key enzymes with various strategies, have been constructed. Based on the evaluation of the trehalose conversion rate from maltodextrins and strain fermentation characteristics, an optimum production strain that co-expressed three extracellular enzymes in one-step fermentation was screened. In the further trehalose production industry, using this engineered strain to produce extracullular three key enzymes in one-step fermentation would simplify the procedure for multi-enzyme preparation and have the potential in reducing production costs.

## 2. Materials and Methods

### 2.1. Bacterial Strains, Plasmids, and Primers

*E. coli* BL21 (DE3) TrxB and pET32a were used for gene expressions of MTSase and MTHase to construct recombinant *E. coli* strains. *B. subtilis* SCK6 and plasmids pMC68 and pHT43 were used for gene expressions of MTSase and MTHase to construct recombinant *B. subtilis* strains, respectively. All of the strains and plasmids used in this study are listed in Appendix A. Moreover, all of the primers designed in this study are listed in Appendix A.

### 2.2. Construction of the Engineered Strains

The encoding genes of MTSase and MTHase from *Arthrobacter ramosus* S34 (GenBank: AB045141.1), *Sulfolobus acidocaldarius* ATCC33909 (GenBank: NC_020246.1) and *Sulfolobus solfataricus* KM1(GenBank: D64128.1) were synthesized by synbio tech Co. Ktd (Suzhou, China) and were inserted into the expression plasmids pET32a, pMC68, and pHT43 using seamless cloning method [22], respectively. The obtained recombinant plasmids were then transformed into *E. coli* BL21 (DE3) TrxB and *B. subtilis* SCK6, respectively. The recombinant strain, which overexpressed pullulanase PulA, was constructed using a genome integration method according to our previous work [23]. The encoding gene of PulA has been previously reported [24] and deposited in NCBI database with the accession NO. HQ844266.1. The co-expression recombinant strains, which expressed MTSase, MTHase, and PulA, were constructed by transforming both recombinant pMC68 and recombinant pHT43 containing MTSase encoding gene or MTHase encoding gene into the recombinant *B. subtilis* strain with one copy of PulA encoding gene integrated at *amyE* position of the *B. subtilis* SCK6 genome.

### 2.3. Fermentation Analysis of Recombinant B. subtilis Strains

The fermentation of the recombinant *B. subtilis* strains for enzyme production was performed according to our previous report [23]. Single colonies were selected after LB plate (25 µg/mL kanamycin and/or 10 µg/mL chloromycetin) activation and inoculated into fresh LB culture liquid with the same antibiotic addition for 14 h of cultivation. The culture broth was then transferred to 30 mL SR medium (30 g/L tryptone, 50 g/L yeast Extract, and 6 g/L K_2_HPO_4_) with 1% inoculum at the related antibiotic addition in a 250 mL shake flask. During 24–72 h cultivation at 37 °C and 220 rpm, the extracellular protein expressions and enzyme activities were detected at fixed intervals. The extracellular supernatant was obtained from the fermentation medium after centrifugation at 12,000 rpm for 10 min. The multiple enzymes solution was prepared and stored via a two-step operation consisting of a filtration with water-based membranes (0.22 μm) and an ultrafiltration concentration by molecular sieving. 

### 2.4. Analysis of Protein Expression and Activity of the Recombinant Enzymes

Extracellular protein expressions were measured using sodium dodecyl sulfate polyacrylamide gel electrophoresis (SDS-PAGE) method reported [25]. Pullulanase activity was measured using the modified dinitrosalicylic acid (DNS) method described in our previous work [25,26]. A mixture composed of 50 μL diluted extracellular pullulanase and 50 μL of pullulan substrates (5% (*w*/*v*)) was added into 400 μL phosphate buffer (pH 6.0) and water bathed at 60 °C for 30 min. Then, the reaction mixture was added with 500 μ L DNS solution and boiled for 10 min. After cooling, its absorbance at OD_540nm_ was measured using microplate spectrophotometer Epoch 2TC (BioTeK, Winooski, VT, USA). One standard pullulanase unit was defined as the amount of enzyme that releases 1 μmol reducing sugar per minute. 

MTSase activity was measured using the 3, 5-dinitrosalicylic acid (DNS) method [27] with minor modifications. A mixture composed of 50 μL diluted extracellular MTSase and 50 μL of maltohexanose (1% (*w*/*v*)) was added into 400 μL phosphate buffer (20 mM, pH 6.0) and water bathed at 50 °C for 30 min. Then, 500 μL DNS solution was added to the reaction liquid, and the mixture was boiled for 10 min. After cooling, its absorbance at OD_540nm_ was measured using microplate spectrophotometer Epoch 2TC (BioTeK, Winooski, VT, USA). One unit (U) of MTSase activity was defined as the amount of enzyme needed to convert 1 μmol of maltohexanose into maltotetraosyltrehalose per min. 

MTHase activity was measured according to a previous report [11] with some modifications. Firstly, maltohexanose was dissolved to a concentration of 1% (*w*/*v*) in 20 mM phosphate buffer (pH 6.0). Then, 50 µL ARS solution was mixed with 0.4 mL maltohexanose solution to produce maltotetraosyltrehalose. After 2 h incubation at 50 °C, the reaction was stopped by heating for 10 min in boiling water. After that, the product solution was incubated at 50 °C. Additionally, 50 µL of appropriately diluted MTHase was added, and the mixture was allowed to react for 30 min. Then, 500 μL DNS solution was added to the reaction mixture and boiled for 10 min. After cooling, the absorbance at OD_540nm_ was measured using microplate spectrophotometer Epoch 2TC (BioTeK, Winooski, VT, USA). One unit (U) of MTHase activity was defined as the amount of MTHase needed to produce 1 µmol of trehalose per min under the assay conditions. 

### 2.5. Enzymatic Trehalose Production from Maltodextrin

For the trehalose production with three-enzyme co-catalysis, each enzyme produced by different engineered strains, the dose of MTSase, MTHase, and pullulanase was 90 U, 65 U, and 0–8 U per gram of maltodextrin. The concentration of the substrate maltodextrin (DE 12) was 10% (wt/vol). The reaction was taken at 50 °C and pH 6.0 for 48 h [28]. For the trehalose production with multi-enzymes produced by the engineered strain in one-step fermentation, 500 μL extracellular enzyme solution was mixed with 500 μL 20% (*w*/*v*) maltodextrin (DE 12) solution, and the mixture was incubated at 50 °C for 24–72 h. The reaction was terminated by incubating in boiling water for 10 min. Commercial glucoamylase (LONCT, Shandong, China) was used to hydrolyze the residual maltodextrin at 60 °C for 24 h. The reaction mixture was then reheated in boiling water for 10 min to inactivate the glucoamylase. The quantity of trehalose was determined via HPLC, according to previous reports [12,29]. 

### 2.6. Determination of the Activity of Mixed Enzymes Obtained via One-Step Fermentation

Maltodextrin (DE 12) was dissolved to a concentration of 20% (wt/vol) in 20 mM phosphate buffer (pH 6.0). Then, 500μL maltodextrin substrate solution was mixed with 500 μL extracellular multi-enzymes solution produced via one-step fermentation. The reaction was carried at 50 °C for 48 h. After reaction, the mixture was boiling for 10 min, and then the cooling solution was centrifuged at 14,000× *g* for 10 min. The supernatant was filtrated with 0.22 μm syringe filters before HPLC analysis. The quantity of product was determined via HPLC using commercial trehalose (Purity ≥ 98%) (Sigma-Aldrich, St. Louis, MO, USA) as standard. The commercial trehalose standard sample was dissolved in double-distilled water with a concentration of 10 mg/mL. The Hypersil NH_2_-S column was used to determine the concentration of trehalose using a refractive index detector (RID). The mobile phase was acetonitrile/water solution (acetonitrile/water = 75:25) with a flow rate of 1.0 mL/min at 40 °C. The quantitative of trehalose in the sample was determined according to the retention time of the standard. The concentration of the trehalose was calculated based on the peak area of the sample. The calculation formula was as $C_{m}=\frac{A_{m}\cdot C_{s}}{A_{s}}\;$ (*C_m_* is concentration of the trehalose, (g/L); *A_m_* is peak area of the sample; *As* is peak area of the standard; *Cs* is the quality of the standard, g). One unit (U) of the activity of the extracellular multi-enzymes produced via one-step fermentation was defined as the amount of enzymes needed to produce 1 µmol of trehalose during 48 h reaction under the assay conditions. The calculation formula was $Activity\;of\;the\;extracellular\;multienzymes=\frac{C_{m}\times Vm\times 2\times 10^{3}}{\mathrm{Molecular}\;\mathrm{weight}\;\mathrm{of}\;\mathrm{trehalose}}\;$ (*V_m_* is total volume of reaction system).

### 2.7. Detected Stability of the Strain’s Enzyme Production and Complex Enzymes

The single colony of *B. subtilis* PSH02 was inoculated into 2 mL of LB liquid medium. After 8 h cultivation, 20 μL bacterial solution was inoculated into 2 mL fresh LB medium and continued to culture for 8 h. After each 8 h interval, the solution was determined as a generation. The 1st, 5th, 10th, 15th, 20th, 25th, and 30th generations were selected for shaking flask fermentation, and then the extracellular multi-enzymes were used to catalyze maltodextrin converting to trehalose. The trehalose conversation rates catalyzed by multi-enzymes from different generation strains were compared to determine the passage stability of *B. subtilis* PSH02. Meanwhile, the extracellular multi-enzymes solution after 24 fermentation of *B. subtilis* PSH02 was taken to be filtrated with 0.22 μm syringe filters. The sterile multi-enzyme solutions were stored at room temperature (25–35 °C) and 4 °C for 14 days. The multi-enzyme solutions were selected at the 1st, 3rd, 5th, 7th, and 14th days to detect the trehalose conversation rates from maltodextrin, respectively. The preservation stability of the multi-enzymes solution was determined using the trehalose conversation rate comparisons. 

## 3. Results and Discussion

### 3.1. Screening of the Malto-Oligosyltrehalose Synthase and Malto-Oligosyltrehalose Trehalohydrolase from Different Sources

Three pairs of malto-oligosyltrehalose synthase (MTSase) and malto-oligosyltrehalose trehalohydrolase (MTHase) from *Arthrobacter ramosus* S34 and *Sulfolobus acidocaldarius* ATCC33909 and *Sulfolobus solfataricus* KM1 were heterologously expressed in *E. coli* BL21 (DE3) trxB, respectively. MTSase (ARS) and MTHase (ARH) from *Arthrobacter ramosus* S34; MTHase (SAH) from *Sulfolobus acidocaldarius* ATCC33909; and MTHase (SSH) from *Sulfolobus solfataricus* KM1 were solubly expressed with plasmid pET32a in the TrxA-fused version (Figure 1a). However, MTSase (SAS) from *Sulfolobus acidocaldarius* ATCC33909 and MTSase (SSS) from *Sulfolobus solfataricus* KM1 were expressed in inclusion body forms (Figure 1a). As only ARS has been expressed in a soluble form showing enzyme activities, we further compared the activities of the recombinant ARH, SAH, and SSH using the reaction products of the recombinant ARS as the reaction substrates. The results showed that the activity of ARH was 7.15 times higher than that of SAH (Figure 1b). Moreover, although SSH showed a similar activity with ARH (Figure 1b), the protein amount of SSH was remarkably higher than that of ARH in the same volume of protein addition (Figure 1). From the results, we could speculate that the key enzymes of the two-step continuous reaction should be from the same source to ensure the best conversion efficiency.

We use the fusion expression tag CBM68, which was developed in our previous study [23], to enhance the secretory expressions of ARS and ARH. To our delight, the recombinant ARS and ARH were detected in the medium after the fermentations of the recombinant *B. subtilis* strains with a high-copy plasmid pMC68 and a low-copy plasmid pHT43-C68, respectively (Figure 2a). Similarly to our previous results [23], both the CBM68 fused enzymes and the untagged enzymes were found in the fermentation medium (Figure 2a). The extracellular activities of the recombinant enzymes showed that the CBM68 fused recombinant ARS and ARH with pMC68 plasmid had relatively higher activities by 21.6 U/mL and 14.1 U/mL, which were 5.17 and 3.19 times higher than those of the recombinant ARS and ARH expressed in *E.coli*, correspondingly (Figure 2b). The relatively lower activities of the CBM68 fused recombinant ARS and ARH in pHT43 plasmids were properly caused by lower protein expression amounts (Figure 2). Meanwhile, these results proved that CBM68 indeed played a significant role in increasing the expression of foreign proteins in *B. subtilis* once again. Therefore, we selected the CBM68 fused recombinant ARS and ARH for the construction of further expression strains in *B. subtilis*.

### 3.2. Neutral Pullulanase PulA Enhanced Trehalose Conversion Rate

In the vitro conversion system, when the additional amount of ARH was set as 65 U/g maltodextrin substrate, the trehalose conversion rate increased with the increase in the additional amount of ARS (Figure 3a). The conversion rate was 33% at the highest ARS addition of 90 U/g limited by experimental conditions. However, at the fixed ARS additional amount of 90 U/g, the optimum addition amount of ARH was 32 U/g, leading to the highest trehalose conversion rate of 35% (Figure 3b). At the basis, a neutral pullulanase PulA was added with different additional amounts of 0 U/g, 1 U/g, 3 U/g, 6 U/g, and 8 U/g, respectively. As shown in Figure 3c, when the additional amount of PulA was 3 U/g, the trehalose conversion rate was up to 86%, which was 2.46 times higher than that without the pullulanase addition. It indicated that PulA was very useful for the enzymatic synthesis of trehalose from maltodextrin. Therefore, we decided to make PulA, ARS, and ARH the target enzymes to be co-overexpressed in *B. subtulis*. Moreover, based on the results of vitro multi-enzymatic conversion, pullulanase supplementation was very important to improve the trehalose conversion rate, but the requirement for pullulanase activity is relatively lower compared with the main enzymes ARS and ARH. Thus, to keep a high trehalose conversion rate, the construction of an engineered strain for one-step fermentation-produced complex enzymes still should ensure the expressions of ARS and ARH as high as possible in the first place.

### 3.3. Construction of the Engineered Strains for Co-Production of the Three Enzymes

For the co-expression of the three enzymes, the PulA encoding gene was inserted at the *amyE* site of the genome of *B. subtilis* SCK6, while the ARS and ARH encoding genes were co-expressed using plasmids pMC68 or pHT43-C68, respectively. Four recombinant strains were successfully constructed to determine the extracellular expression of each target protein and the trehalose conversion rates catalyzed by the extracellular multi-enzymes produced by different strains: *B. subtilis* PSH01 (*amyE*::*pulA*/pMC68-ARS/pHT43-C68-ARH), *B. subtilis* SH01 (pMC68-ARS/pHT43-C68-ARH), *B. subtilis* PSH02 (*amyE*::*pulA*/pHT43-C68-ARS/pMC68-ARH), and *B. subtilis* SH02 (pHT43-C68- ARS/pMC68-ARH). The results were shown in Figure 4 and Appendix A. Although the strains *B. subtilis* PSH01 and *B. subtilis* SH01 showed relatively higher expression amounts of ARS after 48 h of fermentation using the high copy number plasmid pMC68 (Figure 4a), the strains *B. subtilis* PSH02 and *B. subtilis* SH02 showed relatively higher trehalose conversion rates at different fermentation times (Figure 4b). However, because ARH was expressed using pMC68 in the recombinant strains *B. subtilis* PSH02 and *B. subtilis* SH02, the ARH expression levels of *B. subtilis* PSH02 and *B. subtilis* SH02 showed relatively higher rates than those of *B. subtilis* PSH01 and *B. subtilis* SH01 (Figure 4a). It indicated that the over-expression of ARH in the system, when a single strain was used to produce multiple enzymes, was relatively important for the improvement of the trehalose conversion rate. Moreover, when compared with *B. subtilis* SH02, the trehalose conversion rate in *B. subtilis* PSH02 was 1.45 times higher catalyzed by its extracellular multi-enzymes after 24 h fermentation (Figure 4b and Appendix A). It was concluded that the co-expression of PulA in the multi-enzyme expression system played a significant role in improving the trehalose conversion rate. As shown in Figure 4b, the effects of different fermentation times of the four recombinant strains on the trehalose conversion rates were evaluated. According to the results, *B. subtilis* PSH02 could reach the highest trehalose conversion rate after 24 h fermentation, while *B. subtilis* SH02 needed 48 h of fermentation to reach its highest trehalose conversion rate (Figure 4b). Moreover, the highest trehalose conversion rate by *B. subtilis* PSH02 was up to 80%, which was 1.29 times higher than that of *B. subtilis* SH02 (Figure 4b). However, with the extension of fermentation time, the trehalose conversion rates in *B. subtilis* PSH02 declined gradually, which was contrary to the increases in the extracellular expressions of target proteins (Figure 4b,c). As the cells grew and plasmids replications, the ARH was expressed and accumulated into a high but unmatched amount to the ARS and substrate, which led to the hydrolysis of the substrate and the reduced trehalose conversion rate accordingly (Appendix A). This unconventional hydrolysis was also proved by the partial increase in trehalose yielding during the determinations of the PulA lacking strain SH02, as shown in Figure 4b. ARH caused the hydrolysis of some α-1,4-glycosidic bonds of maltodextrin, and more reducing ends were revealed to be new substrates, increasing the trehalose production temporarily. With the prolongation of the fermentation time, this enhancement was limited due to the unmatched expression level of ARS and ARH, which caused the decreasing trend of trehalose production ultimately. It indicated that compared with increasing the expression level of each target protein, keeping the appropriate ratios of the three enzymes was more important for enhancing the trehalose conversion rate catalyzed via this multi-enzymatic system produced using one-step fermentation. As shown in Figure 4, at the fermentation time of 24 h, *B. subtilis* PSH02, producing the optimum ratios of ARS, ARH, and PulA in the extracellular medium made the trehalose conversion rate up to 80% from 10 g/L maltodextrin during 48 h of reaction at 50 °C.

The engineered strains, which were inserted with ARS and/or ARH into the genome of *B. subtilis* SCK6, were also constructed in this work. Firstly, ARS and ARH were separately inserted into the *amyE*, *nprB*, or *ytxE* position of the genome of *B. subtilis* SCK6 to test the enzyme activity. The results are shown in Appendix A, the same enzyme which showed different activities at different inserted positions. Similarly, ARS and ARH showed remarkably different enzyme activity values when they were inserted into the same positions. However, all recombinant ARS and ARH, which were expressed by genome integration, showed activities of no more than 5 U/mL (Appendix A). Moreover, the co-expressions of ARS and ARH in different combination sites also showed different trehalose conversion rates, and the highest trehalose conversion rate was only 26% (Appendix A). Therefore, compared with the engineered strains which over-expressed ARS and ARH by plasmids, the genome integration strains showed relatively lower enzyme activities and were not suitable to be multi-enzyme-producing strains for the production of trehalose.

### 3.4. Properties of the Engineered Strain B. subtilis PSH02 for Trehalose Enzymatic Synthesis

Because various enzyme production ratios of ARS, ARH, and PulA would lead to different trehalose conversion rates, in order to evaluate the transformation ability of the multi-enzymes produced by the engineered strains for trehalose production, the multi-enzyme activity was defined as the number of enzymes that produce 1 μM trehalose during 48 h of reaction at 50 °C. The results showed that the multi-enzymes activities produced by *B. subtilis* SH01, *B. subtilis* PSH01, *B. subtilis* SH02, and *B. subtilis* PSH02 were 23.2 U/mL, 36.5 U/mL, 38.4 U/mL, and 51.7 U/mL after 24 h fermentation, correspondingly. The multi-enzymes activity of each strain corresponded to its trehalose conversion rate very well (Figure 4b). On this basis, we used the extracellular multi-enzymes produced by *B. subtilis* PSH02 after 24 h of fermentation to further optimize the substrate concentration and enzyme dosage. As shown in Figure 5a, 15 U/mL of the multi-enzymes could catalyze more than 80% of 10 g/L maltodextrin substrate to trehalose. When the concentration of maltodextrin substrate was 100 g/L, the additional dosage of the multi-enzymes needed to be 150 U/mL to reach the eligible trehalose conversion rate, which was more than 80% (Figure 5b). It illustrated that the three times concentration of the extracellular multi-enzymes produced by *B. subtilis* PSH02 after 24 h if fermentation could obtain more than 80% of trehalose conversion rate with the high substrate concentration of 100 g/L. Compared with the work carried out by Liu, who utilized the recombinant *Bacillus subtilis* to secrete TreS and converted maltose to trehalose with a conversion rate of 75.5% [2], a higher conversion rate was obtained during our work as the engineered strain *B. subtilis* PSH02 showed a good potential for industrial application of trehalose enzymatic synthesis.

In order to further evaluate the production characteristics of *B. subtilis* PSH02, the passage stability of *B. subtilis* PSH02 and the stability of its extracellular multi-enzyme activity were determined in this work. As shown in Figure 6a, the relative activity of the extracellular multi-enzymes of *B. subtilis* PSH02 was higher than 88% after a successive passage to 30 generations. It indicated that *B. subtilis* PSH02 had a high production stability of the extracellular multi-enzymes. Moreover, both at 4 °C and room temperature (25–35 °C), the extracellular multi-enzymes could keep high relative activities during 7 days of preservation (Figure 6b). After 14 days of preservation at 4 °C and room temperature (25–35 °C), the relative activities of the extracellular multi-enzymes were 82% and 83%, respectively. Thus, the stabilities, both in *B. subtilis* PSH02 generations and its extracellular multi-enzymes activities, made *B. subtilis* PSH02 a potential industrial production strain.

## 4. Conclusions

In this study, four engineered strains were constructed to produce the key enzymes via one-step fermentation for trehalose enzymatic synthesis from maltodextrin: *B. subtilis* PSH01 (*amyE*::pulA/pMC68-ARS/pHT43-C68-ARH), *B. subtilis* SH01 (pMC68-ARS/pHT43-C68-ARH), *B. subtilis* PSH02 (*amyE*::pulA/pHT43-C68-ARS/pMC68-ARH) and *B. subtilis* SH02 (pHT43-C68- ARS/pMC68-ARH). The optimum strain *B. subtilis* PSH02, which co-expressed ARS, ARH, and PulA at an appropriate ratio after 24 h fermentation, could make the highest trehalose conversion rate of 80% from 100 g/L maltodextrin substrate. Moreover, *B. subtilis* PSH02 showed high generational stability within 30 generations. Its extracellular multi-enzymes could keep high preservation stability within 7 days at room temperature without adding any protective agent. Thus, the engineered strain *B. subtilis* PSH02 showed a high prospect for industrial application in trehalose enzymatic synthesis.

## Figures and Tables

**Figure 1 bioengineering-10-00977-f001:**
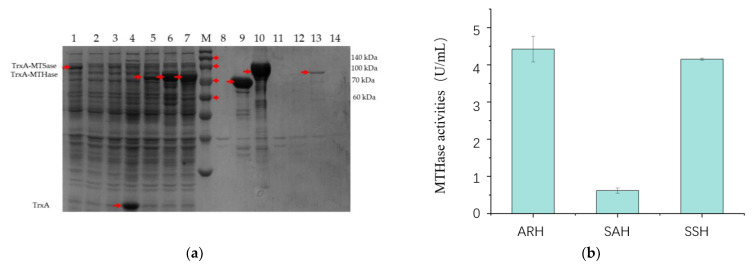
Heterologous expressions of MTSase and MTHase in *E. coli* BL21 (DE3) trxB. (**a**) SDS-PAGE of the intracellular soluble and insoluble proteins of the recombinant *E. coli* strains. Lane M: protein molecular weight standards. Lanes 1 and 8: the intracellular soluble and insoluble proteins of the recombinant *E. coli* strains expressing recombinant ARS; lane 2 and 9: the intracellular soluble and insoluble proteins of the recombinant *E. coli* strains expressing recombinant SAS; lane 3 and 10: the intracellular soluble and insoluble proteins of the recombinant *E. coli* strains expressing recombinant SSS; lane 4 and 11: the intracellular soluble and insoluble proteins of the control strains with the empty plasmid pET32a in *E. coli* BL21(DE3) trxB; lane 5 and 12: the intracellular soluble and insoluble proteins of the recombinant *E. coli* strains expressing recombinant ARH; lane 6 and 13: the intracellular soluble and insoluble proteins of the recombinant *E. coli* strains expressing recombinant SAH; lane 7 and 14: the intracellular soluble and insoluble proteins of the recombinant *E. coli* strains expressing recombinant SSH. (**b**) MTHase activities of ARH, SAH, and SSS using the reaction products of ARS as the reaction substrate. All values in the histogram are expressed in means ± SD (*n* = 3).

**Figure 2 bioengineering-10-00977-f002:**
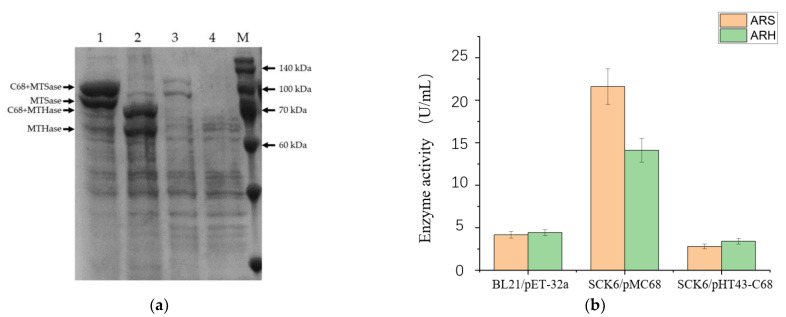
Heterologous expressions of ARS and ARH in *B. subtilis* SCK6. (**a**) SDS-PAGE of the extracellular proteins of the recombinant *B. subtilis* strains. Lane 1: ARS expressed in *B. subtilis* SCK6 using the expression plasmid pMC68; lane 2: ARH expressed in *B. subtilis* SCK6 using the expression plasmid pMC68; lane 3: ARS expressed in *B. subtilis* SCK6 using the expression plasmid pHT43 (N-terminal fused with CBM68); lane 4: ARH expressed in *B. subtilis* SCK6 using the expression plasmid pHT43 (N-terminal fused with CBM68); lane M: protein molecular weight standards. (**b**) The activities of the recombinant enzymes are heterologously expressed by the engineered strains. All values in the histogram are expressed in means ± SD (*n* = 3).

**Figure 3 bioengineering-10-00977-f003:**
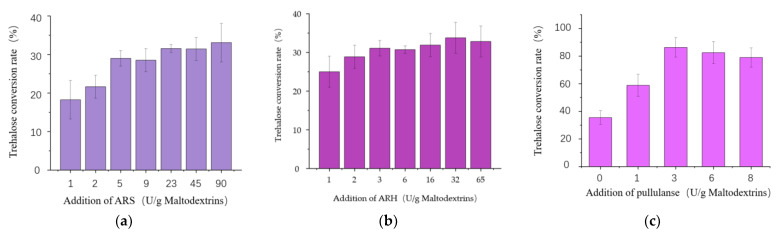
Effect of different enzyme additions on trehalose conversion rate. (**a**) The optimum additional amount of ARS for trehalose production. (**b**) The optimum additional amount of ARH for trehalose production. (**c**) The optimum additional amount of PulA for trehalose production. All values are expressed as the means ± SDs (*n* = 3).

**Figure 4 bioengineering-10-00977-f004:**
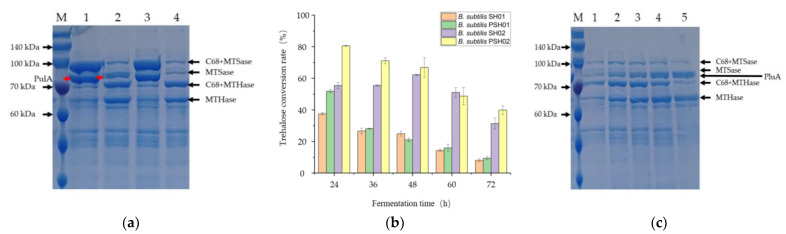
Co-expression of ARS, ARH and PulA in *B. subtilis* SCK6. (**a**) SDS-PAGE of the extracellular proteins of the co-expression strains. Lane M: protein molecular weight standards; lane 1: extracellular proteins of the recombinant strain *B. subtilis* PSH01 (*amyE*::*pulA*/pMC68-ARS/pHT43-C68-ARH) after 48 h fermentation; lane 2: extracellular proteins of the recombinant strain *B. subtilis* PSH02 (*amyE*::*pulA*/pHT43-C68-ARS/pMC68-ARH) after 48 h fermentation; lane 3: extracellular proteins of the recombinant strain *B. subtilis* SH01 (pMC68-ARS/pHT43-C68-ARH) after 48 h fermentation; lane 4: extracellular proteins of the recombinant strain *B. subtilis* SH02 (pHT43-C68-ARS/pMC68-ARH) after 48 h fermentation. (**b**) Trehalose conversion rates using the extracellular enzymes produced by the co-expression strains after different fermentation times. (**c**) SDS-PAGE of the extracellular proteins of *B. subtilis* PSH02 with different fermentation times. lane 1: 24 h; lane 2: 36 h; lane 3: 48 h; lane 4: 60 h; lane 5: 72 h. All values in the histogram are expressed in means ± SD (*n* = 3).

**Figure 5 bioengineering-10-00977-f005:**
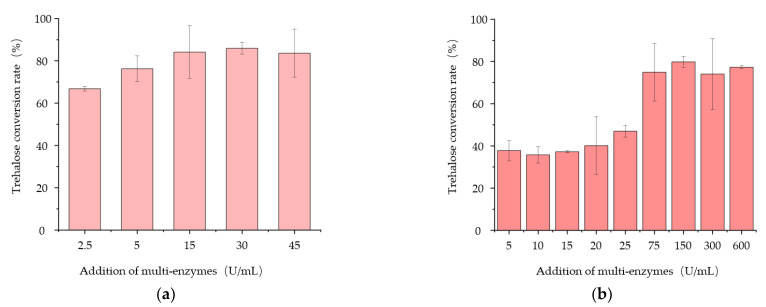
Effects of different addition dosages of multi-enzymes on the trehalose conversion rate at the maltodextrin substrate concentration of 10 g/L (**a**) and 100 g/L (**b**). All values are expressed as the means ± SDs (*n* = 3).

**Figure 6 bioengineering-10-00977-f006:**
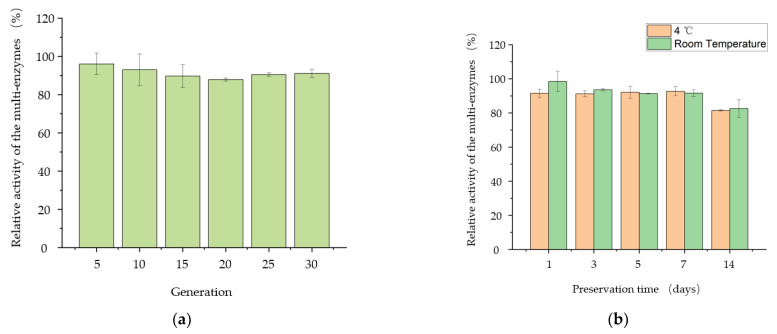
Evaluation of the production characteristics from *B. subtilis* PSH02. (**a**) Passage stability of *B. subtilis* PSH02. The activity of extracellular multi-enzymes produced by the original generation *B. subtilis* PSH02 after 24 h fermentation was set as 100%. (**b**) Stability of its extracellular multi-enzymes activity. The activity of the fresh extracellular multi-enzymes produced by *B. subtilis* PSH02 after 24 h fermentation was set as 100%. All values are expressed as the means ± SDs (*n* = 3).

## Data Availability

The data presented in this study are available in the text of the article and Appendix A. Further inquiries can be directed to the corresponding author.

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
