# Peer review of "Trehalose Production Using Three Extracellular Enzymes Produced via One-Step Fermentation of an Engineered Bacillus subtilis Strain"

_bioengineering, 2023, doi:10.3390/bioengineering10080977_

Round 1
Reviewer 1 Report
In this manuscript, authors showed an approach of trehalose production using three extracellular enzymes produced by one-step fermentation of an engineered Bacillus subtilis. Most of the conclusions of this manuscript were supported by the data. The manuscript has potential interest. However, several comments need to be addressed before publication.
Major comments:
1. I think the author should include a schematic diagram to show the metabolic pathway (or proposed pathway) of converting maltodextrin to trehalose by MTSase, MTHase and PulA.
2. How did the authors get the extracellular multi-enzymes solution after the fermentation? Were there any isolation and purification processes included? The authors should make it clear in the Materials and Methods section.
3. Why the authors didn’t use high copy number plasmids for the expression of both of ARS and ARH?
4. Line 254-256, authors said that the optimum addition amount of ARS was 90 U/g, but Fig. 3(a) didn’t show the data of adding ARS more than 90 U/g. Also, it seemed that Fig. 3(b) could not show the optimum addition amount of ARH was 32 U/g.
5. The authors claimed B. subtilis PSH02 produced the optimum ratios of ARS, ARH and PulA in the extracellular medium at the fermentation time of 24 h and keeping appropriate ratios of the three enzymes were important for enhancing the trehalose conversion rate catalyzed by this multi-enzymatic system. However, I don’t think the protein gel showed in Fig. 4c can be evidence to support this conclusion. What’s this optimum ratio at 24 h and what are the ratios at other fermentation times? Why do different ratios of these three enzymes lead to different trehalose conversion rates? I would think this conclusion is just a speculation by the authors and there is no direct evidence to support this conclusion. Is it possible that there were some metabolites included in the extracellular multi-enzymes solution as the fermentation time increases and inhibited the conversion of trehalose? I think there could have many possible reasons for the reduction of trehalose conversion rates as the fermentation time increases which should be fully discussed. If the authors thought different ratios of these three enzymes could be the reason, the authors also need to completely discuss and explain.
6. I think the authors should add some HPLC chromatograms in the SI.
Minor comments:
1. Line 33-34, I don’t think the authors could claim this approach would greatly reduce the production costs without any data and comparison with other work.
2. I think the authors should add some content in the Introduction section to explain the motivation of using B. subtilis as the host to express this multi-enzymatic system.
3. Can the authors add maltodextrin in the fermentation medium to directly produce trehalose during fermentation?
4. Line 66-67, I think there should be some references.
5. Line 231-232, the authors named pHT43-CBM68 in the text, but pHT43-C68 in the Fig. 2b. They should be consistent.
6. Some small writing and grammar issues should be corrected.
Line 39, “was” should be “is”
Line 58, “was” should be “is”
Line 62, “making” should be “have made”
Line 65, “was” should be “is”
Line 69, “were” should be “are”
Line 209, “speculated” should be “speculate”
Line 234-235, I guess the sentence should be “the CBM68 fused recombinant ARS and ARH in pMC68 plasmid had the relatively higher activities with 21.6 U/ml and 14.1 U/ml”.
Line 261, “is” should be “was”.
Line 282, should be “As the results shown in Fig. 4”
The manuscript is well written, but minor editing of English language is required.
Reviewer 2 Report
The manuscript titled “Trehalose production using three extracellular enzymes produced by one-step fermentation of an engineered Bacillus subtilis strain” describes the one-step production of trehalose using fermentation of B. subtilis. The results may be fine; however, the presentation of the manuscript is very poor. It is very poorly written.
Due to several spelling mistakes, grammatical mistakes and mistakes in sentence making, the manuscript is difficult to understand. Kindly consult some expert and correct the manuscript to make it understandable.
Comment 1. Line 148: How were the reaction conditions optimized to 50 degrees and pH 6.5?
Comment 2. What is the significance of Line 165?
Comment 3. Line 188: What is the shelf life of the multi- enzyme at room temperature? Please give one value of the room temperature (25-35 degrees?)?
Comment 4. Please check the use of the word “respectively” in the manuscript (Particularly in section 3.1 and 3.2).
Comment 5. Figure 1, 2 and 4: Please refer to ‘band’ as ‘lane’ in the legend.
Comment 6. Line 252: Please check “In the vitro transformation system”?
Comment 7. Please compare your results with the previous studies. Please discuss the significance of the results.
Comment 8. How does pullulanase addition increase the trehalose production?
Comment 9. Figure 6: In the legend, please describe Figure a and b in continuity.
Very poor
Round 2
Reviewer 1 Report
The authors addressed all of my comments. I am satisfied with the current version. I recommend accepting the manuscript in present form.
Reviewer 2 Report
Authors have significantly improved the manuscript.